# Unraveling the Complexity of Chikungunya Virus Infection Immunological and Genetic Insights in Acute and Chronic Patients

**DOI:** 10.3390/genes15111365

**Published:** 2024-10-24

**Authors:** Hegger Fritsch, Marta Giovanetti, Luan Gaspar Clemente, Gabriel da Rocha Fernandes, Vagner Fonseca, Maricelia Maia de Lima, Melissa Falcão, Neuza de Jesus, Erenilde Marques de Cerqueira, Rivaldo Venâncio da Cunha, Marcos Vinicius Lima de Oliveira Francisco, Isadora Cristina de Siqueira, Carla de Oliveira, Joilson Xavier, Jorge Gomes Goulart Ferreira, Fábio Ribeiro Queiroz, Elise Smith, Jennifer Tisoncik-Go, Wesley C. Van Voorhis, Peter M. Rabinowitz, Judith N. Wasserheit, Michael Gale, Ana Maria Bispo de Filippis, Luiz Carlos Junior Alcantara

**Affiliations:** 1Instituto de Ciências Biológicas, Universidade Federal de Minas Gerais, Belo Horizonte 31270-901, Brazil; hegger.fritsch@univ-tours.fr (H.F.); joilsonxavier@live.com (J.X.); 2Institut National de la Santé et de la Recherche Médicale, U1259—MAVIVHe, Université de Tours, 37032 Tours, France; 3Department of Science and Technologies for Sustainable Development and One Health, Università Campus Bio-Medico di Roma, 00128 Rome, Italy; giovanetti.marta@gmail.com; 4Instituto René Rachou, Fundação Oswaldo Cruz, Belo Horizonte 30190-002, Brazil; fernandes.gabriel@gmail.com; 5Climate Amplified Diseases and Epidemics (CLIMADE)—CLIMADE Americas, Belo Horizonte 30190-002, Brazil; 6Escola Superior de Agricultura Luiz de Queiroz, Departamento de Zootecnia, Universidade de São Paulo, Piracicaba 13418-900, Brazil; luan.clemente@usp.br; 7Departamento de Ciências Exatas e da Terra, Universidade Estadual da Bahia, Salvador 41150-000, Brazil; vagner.fonseca@gmail.com; 8Centre for Epidemic Response and Innovation (CERI), School of Data Science and Computational Thinking, Stellenbosch University, Stellenbosch 7600, South Africa; 9Departamento de Saúde, Universidade Estadual de Feira de Santana, Feira de Santana 44036-900, Brazil; limapfa6@gmail.com (M.M.d.L.); eremarques@fsonline.com.br (E.M.d.C.); 10Secretaria de Municipal de Saúde de Feira de Santana, Divisão de Vigilância Epidemiológica, Feira de Santana 44027-010, Brazil; melissa.falcao@hotmail.com (M.F.); neuzaenfajesus@hotmail.com (N.d.J.); 11Fundação Oswaldo Cruz, Bio-Manguinhos, Rio de Janeiro 21040-360, Brazil; rivaldo.cunha@fiocruz.br; 12Instituto Gonçalo Moniz, Fundação Oswaldo Cruz, Salvador 40296-710, Brazil; marcosv19@live.com (M.V.L.d.O.F.); isadora.siqueira@fiocruz.br (I.C.d.S.); 13Laboratório de Arbovírus e Vírus Hemorrágicos, Instituto Oswaldo Cruz, Fundação Oswaldo Cruz, Rio de Janeiro 21040-360, Brazil; oliveirasc@yahoo.com.br; 14Núcleo de Ensino, Pesquisa e Inovação, Instituto Mário Penna, Belo Horizonte 30380-420, Brazil; jorge.ferreira@mariopenna.org.br (J.G.G.F.); fabio.ribeiro@mariopenna.org.br (F.R.Q.); 15Department of Immunology, University of Washington, Seattle, WA 98109, USA; eliser2@uw.edu (E.S.); tisoncik@uw.edu (J.T.-G.); mgale@uw.edu (M.G.J.); 16Department of Medicine, University of Washington, Seattle, WA 98195, USA; wesley@uw.edu; 17Departments of Environmental and Occupational Health Sciences, University of Washington, Seattle, WA 98195, USA; peterr7@uw.edu; 18Department of Global Health, University of Washington, Seattle, WA 98105, USA; jwasserh@uw.edu

**Keywords:** Chikungunya virus, RNA-sequencing, acute infection, Chikungunya chronicity, immune response, immunological characterization

## Abstract

**Background**: The chikungunya virus (CHIKV), transmitted by infected Aedes mosquitoes, has caused a significant number of infections worldwide. In Brazil, the emergence of the CHIKV-ECSA genotype in 2014 posed a major public health challenge due to its association with more severe symptoms. **Objectives/Methods**: This study aimed to shed new light on the host immune response by examining the whole-blood transcriptomic profile of both CHIKV-acute and chronically infected individuals from Feira de Santana, Bahia, Brazil, a region heavily affected by CHIKV, Dengue, and Zika virus epidemics. **Results**: Our data reveal complex symptomatology characterized by arthralgia and post-chikungunya neuropathy in individuals with chronic sequelae, particularly affecting women living in socially vulnerable situations. Analysis of gene modules suggests heightened metabolic processes, represented by an increase in NADH, COX5A, COA3, CYC1, and cap methylation in patients with acute disease. In contrast, individuals with chronic manifestations exhibit a distinct pattern of histone methylation, probably mediated by NCOA3 in the coactivation of different nuclear receptors, KMT2 genes, KDM3B and TET2, and with alterations in the immunological response, majorly led by IL-17RA, IL-6R, and STAT3 Th17 genes. **Conclusion**: Our results emphasize the complexity of CHIKV disease progression, demonstrating the heterogeneous gene expression and symptomatologic scenario across both acute and chronic phases. Moreover, the identification of specific gene modules associated with viral pathogenesis provides critical insights into the molecular mechanisms underlying these distinct clinical manifestations.

## 1. Introduction

Emerging and re-emerging viral infectious diseases impose a significant burden on public health and the global population, leading to devastating outbreaks and affecting a large number of individuals worldwide [1,2]. Among these viruses, arthropod-borne viruses (arboviruses) present particularly alarming scenarios due to their widespread distribution, posing a risk of infection to populations in more than 110 countries across Asia, Africa, Europe, and the Americas [3]. Chikungunya virus (CHIKV) is an arthrogenic alphavirus belonging to the Togaviridae family, transmitted by infected Aedes ssp. mosquitoes [4,5,6,7]. It can be classified into four distinct lineages or genotypes based on genomic differences: (i) the West African lineage; (ii) the East/Central/South African (ECSA) lineage; (iii) the Asian lineage; and (iv) the Indian Ocean lineage (IOL) [8,9,10]. Although there is no consensus in the literature regarding the impact of the viral lineage on the severity and prognosis of the disease, CHIKV-ECSA infections seem to be associated with more severe symptoms compared to other lineages [11]. Since its initial cases, reported in Tanzania in 1953, CHIKV infections have been reported worldwide [3,9,12,13], with almost 4 million probable cases in the Americas alone from 2013 to May 2024 [3,13]. In Brazil, the most affected country on the continent [3,13], the local introduction of the CHIKV-ECSA genotype was first detected in the municipality of Feira de Santana, Northeast region, in 2014. Since then, the ECSA genotype has been reported in several states across the country, posing a serious threat to public health [14,15,16,17,18,19,20,21,22,23], given the uncertain impact of CHIKV-ECSA infection in more severe cases [11].

The vast majority of Chikungunya-infected patients may experience an asymptomatic infection [24]. However, for those who show clinical manifestations, the acute phase may be marked by a sudden onset of fever, followed by cutaneous manifestations (rash), fatigue, and debilitating polyarthralgia. Although most symptoms resolve within a few weeks and the mortality rate for CHIKV is extremely low (estimated to be approximately 0.1%) [24], the persistence of joint symptoms for months or years in a significant proportion of the infected population presents a substantial global economic and public health challenge [25,26]. Presently, there is no highly effective drug or licensed vaccine available to prevent CHIKV complications. Treatment mainly involves the use of paracetamol (acetaminophen) and nonsteroidal anti-inflammatory drugs to alleviate rheumatological symptoms [27,28,29].

Even though several studies have focused on characterizing the immune response to CHIKV infection and describing the immunobiological mechanisms, the natural history of the disease and factors associated with chronicity remain poorly understood [2,30,31,32,33]. They suggest elevated host expression of proinflammatory mediators, such as interleukin 6 (IL-6), monocyte chemoattractant protein 1 (MCP-1), interferon (IFN) α, and IFN-γ, aimed at controlling the pathogen during the acute phase of CHIKV infection [34,35,36,37]. Additionally, IL-6 and MCP-1 have been associated with high viral load (HVL) [38,39], while markers such as interleukin 1β have been correlated with disease severity [40]. Furthermore, studies also suggest that the presence of IL-6 and granulocyte-macrophage colony-stimulating factor (GM-CSF) may be linked with persistent arthralgia [38]. However, further investigation is needed to fully characterize the mediators and pathways associated with the acute and chronic phases of CHIKV infection.

To further elucidate virus-host mechanisms, transcriptomic approaches offer a promising means of accessing molecular and immunological profiles in disease cases. With next-generation sequencing of transcripts, it is possible to investigate and characterize specific signatures in diseases, generating new insights into therapeutics and clinical management [41,42,43]. Recent studies have reported the use of peripheral blood as an alternative to tissue samples for transcriptome studies [42,44], offering the possibility of detecting pathological changes in different phases of the disease without invasive collection. In this study, we aimed to provide new insights into the molecular mechanisms and possible gene modules involved in different phases of Chikungunya infection. For that, we used transcriptomics to characterize acute and chronic CHIKV infection in individuals from Feira de Santana, Bahia, Brazil—the municipality where ECSA-CHIKV was introduced into the country and served as a source of transmission to several regions heavily impacted by the cocirculation of Dengue and Zika viruses.

## 2. Materials and Methods

### 2.1. Human Ethics Statement

This research and recruitment locations were approved by the University of Washington IRB Committee (protocol: STUDY00009300), the Ethical Committee of the Pan American World Health Organization (No. PAHO-2016-08-0029), the Oswaldo Cruz Foundation Ethics Committee (CAAE: 45279715.8.0000.0040), the Federal University of Minas Gerais Ethics Committee (CAAE: 32912820.6.1001.5149), and the Brazilian Ministry of Health (MoH) as part of the arbovirus genomic surveillance efforts. All patients consented to participate by signing a consent form.

### 2.2. Study Design and Sample Collection

To characterize the immunological response across distinct phases of Chikungunya virus infection, we selected 19 cases of chronic CHIKV infection and 13 individuals exhibiting acute symptoms. Individuals with chronic infection ranged in age from 42 to 86 years, while acute cases ranged from 18 to 84 years. The participants were recruited in collaboration with the epidemiological surveillance department of the Health Secretary of Feira de Santana-BA, Brazil, from April to September 2023.

Acute cases suspected of Chikungunya infection underwent nucleic acid extraction and purification using the Reliaprep Viral TNA kit (Promega, Madison, WI, USA) and subsequent laboratory confirmation through multiplex RT-qPCR assay targeting Zika, Dengue, and Chikungunya viruses (ZDC molecular kit, Bio-Manguinhos, Rio de Janeiro, Brazil). Individuals who tested positive for the Chikungunya virus up to 5 days after the onset of symptoms were invited to participate. Patients co-infected with other arbovirus pathogens in the arbovirus diagnosis panel were excluded from the analysis to mitigate potential biases in immunological response assessment.

As an inclusion criterion for the chronic group, all chronic individuals who demonstrated, by clinical report and physical exam, the persistence of symptoms for one or more years following CHIKV infection enrolled. Additionally, individuals with chronic diseases should have been under medical supervision since the convalescent phase (approximately 90 days post-symptom onset) in order to retrieve the medical report over the years. To control for genetic and behavioral biases, healthy individuals genetically related to chronic cases (members of the same family), who had a previous history of CHIKV infection but did not develop chronic disease, were selected as a control group.

A single blood sample was collected from all patients and control individuals in Tempus tubes (Thermo Fisher, Waltham, MA, USA) to stabilize RNA at the time of enrollment. Samples were stored at −20 degrees Celsius at the Experimental Pathology Laboratory (LAPEX)—Gonçalo Moniz Institute/Fiocruz Bahia, Salvador, Brazil, before being sent to the Laboratory of Mosquito Vectors and Endosymbionts at Instituto René Rachou/Fiocruz Minas Gerais, Belo Horizonte, Brazil. Additionally, all patients and control group members underwent clinical interviews and examinations by a physician, during which a clinical-epidemiological questionnaire was administered.

### 2.3. RNA Extraction and mRNA Library Preparation

For total RNA extraction, we used an in-house phenol-chloroform protocol, followed by purification with the RNeasy MinElute Clean-up kit (QIAGEN, Hilden, Germany). For that, 500 μL of the whole blood sample collected in Tempus Blood RNA Tube tubes (Thermo Fisher, Waltham, MA, USA) for RNA stabilization was used to carry out the extraction as described in Protocol S1 and eluted in 26 μL of nuclease-free water. The RNAs obtained were quantified using the Qubit^®^ 3.0 fluorometer and Qubit RNA Quantification High Sensitivity (HS) kit, capable of quantifying within the range of 4–200 ng. After quantification, the RNA integrity score (RIN) was checked using the RNA ScreenTape TapeStation system kit (Agilent, Santa Clara, CA, USA).

After accessing the RNA integrity (RIN), the RNA library preparation procedure was carried out using the Illumina Stranded mRNA Prep Ligation kit (Illumina, San Diego, CA, USA). During the library preparation, the fraction corresponding to messenger RNAs (mRNA) contained in the total RNA fraction was segregated and used for cDNA synthesis with polyadenylated 5′ end tail specific primers. Each sample was identified using a distinct index from the Illumina RNA UD Indexes Ligation index kit (Illumina, San Diego, CA, USA), and the sequencing was performed on the NextSeq 2000 platform with a 300-cycle P3 cartridge for up to 72 h. The protocol was carried out on the Plataforma Genômica–Sequenciamento de Nova Geração–RPT01J (Rede de Plataformas Tecnológicas FIOCRUZ) in accordance with what is stated in the protocol.

### 2.4. Quality Control and Transcriptome Reconstruction

Quality control and adapter removal were performed using the Trim Galore program (version 0.6.7) [45]. Parameters were set to exclude reads below 70 base pairs in length and a Phred quality score (Q score) equal to or below 20. Following quality control, the trimmed reads were aligned to the reference genome using the STAR (Spliced Transcripts Alignment to a Reference) program (version 2.7.11) [46]. The alignment process utilized default parameters provided by the STAR program using the genome assembly GRCh38 retrieved from Ensemble (https://www.ensembl.org/Homo_sapiens/Info/Index, accessed on 15 January 2024) as a reference.

After reading the mapping, transcriptome reconstruction and quantification were performed using the StringTie software (version 2.2.1) [47]. The reconstruction process utilized the Gencode version 45 annotation file, which provided comprehensive gene annotations and transcript structures for the reference genome.

The gffcompare utility was employed to establish the transcriptome’s quality by comparing the reconstructed transcripts obtained in StringTie with the reference annotations provided in the Gencode v. 45 annotation file. This comparative analysis enabled the identification of novel transcripts and the assessment of transcript structure accuracy.

### 2.5. Differential Expression Analysis

Before performing the differential expression analysis, the StringTie output was processed using the prepDR.py3 script available on the StringTie website. To identify differentially expressed genes across various stages of Chikungunya infection, we utilized the edgeR package (version 4.0.16) [48] in the R statistical environment. The workflow included normalizing read counts, estimating dispersion parameters, and fitting statistical models to detect genes showing significant changes in expression between conditions.

Dispersion estimation in edgeR [48] was conducted in three stages. First, common dispersion was estimated using the estimateGLMCommonDisp function, which calculates a single dispersion value for all genes, assuming shared variability across samples. Next, trend dispersion was assessed using the estimateGLMTrendedDisp function to capture any relationship between gene abundance and variability. Finally, gene-specific (tagwise) dispersion was estimated using the estimateGLMTagwiseDisp function, providing individual dispersion estimates for each gene.

Statistical models were then fitted using the glmFit function, which applies a negative binomial generalized log-linear model (GLM) to the read count data. To identify significant differences in gene expression between conditions, we used the likelihood ratio test implemented in the glmLRT function. A contrast matrix was constructed to specifically compare gene expression between control versus acute and control versus chronic infection stages.

To ensure the reliability of the results, the false discovery rate (FDR) was controlled using the Benjamini–Hochberg correction. Genes were considered differentially expressed if they showed an FDR of <0.05 and a log fold-change (logFC) greater than |1|.

### 2.6. Gene Coexpression and Functional Enrichment Pathway Analysis

Following the differential expression analysis, we utilized the Weighted Gene Coexpression Network Analysis (WGCNA) package (version 1.72-5) [49] in R to identify gene coexpression modules. Prior to this, the gene expression data were converted to log2 counts per million (log2CPM) to ensure accurate comparisons. We employed the sft_fit function to determine the soft-thresholding power for network construction, utilizing a signed hybrid network type to capture both positive and negative correlations. Pearson correlation was used to calculate the correlation coefficients.

Subsequently, we correlated the identified modules with the various stages of Chikungunya infection using the module_trait_cor function. Modules exhibiting statistically significant correlations (*p*-value < 0.05) were prioritized for further investigation.

To elucidate the biological functions and pathways associated with the identified coexpression modules, we performed functional enrichment analysis using the gProfiler tool (available at https://biit.cs.ut.ee/gprofiler/, accessed on 15 February 2024) [50,51] and STRING (available at https://string-db.org/, accessed on 15 February 2024). Enriched terms were considered biologically relevant if they had a *p*-value < 0.05 and a significant number of genes involved in the pathway.

For further exploration of the significantly correlated modules, we applied the Markov Cluster Algorithm (MCL) [52] on STRING, using a graph clustering method based on stochastic flow simulation. An inflation parameter of 3 was used, and the resulting clusters within each module were visualized using the STRING platform (available on https://string-db.org/, accessed on 15 February 2024).

## 3. Results

### 3.1. Social Vulnerability Marks Chikungunya Infection in Chronic Patients

Among the sampled individuals, we observed a significant predominance of female patients, representing 89.6% of the chronic population (Table 1), while the control population was predominantly male. Both the chronic and control groups had a similar average age of 60.4 and 57.8 years, respectively, while individuals in the acute phase were younger with an average age of 44 years.

Regarding the socioeconomic factors, our analysis revealed that the chronic population predominantly (94.8%) (Table 1) consisted of individuals from families with a per capita income of up to twice the minimum monthly wage (minimum monthly wage of 1412.00 BRL). This result aligns with the observed low level of education within this population, with nine individuals having incomplete secondary education and only 10.6% possessing a higher education level (Table 1).

### 3.2. A Complex Symptomatological Scenario

The most prevalent symptoms, persisting into the chronic phase, included arthralgia (100%), with a notable occurrence of bilateral involvement (100%), edema (84.8%), in addition to myalgia (42.4%), transient arthralgia, and prostration (31.8% and 26.5%, respectively). The sites most affected by arthralgia were regions of the appendicular system, such as wrists (84.8%), fingers and toes (68.9% and 63.6%, respectively), ankles (74.2%), and knees (89.6%). Continued discomfort in axial skeleton regions, including the neck (26.5%) and lumbar spine (31.8%), was also reported (Figure 1A). Similar patterns were observed for myalgia, with 26.5% of individuals reporting muscular discomfort in the arms, shoulders, and calf regions, and 21.2% and 31.8% indicating the arms and thighs as the most affected regions, respectively (Figure 1B).

In acute cases, fever and myalgia were unanimously reported, followed by arthralgia (92.3%). Skin symptoms were reported in 53.8% of individuals, along with gastrointestinal symptoms such as nausea (61.6%). Additionally, 69.2% of acute cases reported migraine, while 53.8% presented retro-orbital pain. Focal points of myalgia and arthralgia were not adequately documented for ten individuals (76.9%).

Considering the potential impact of medications on immune response outcomes, medication intake at the time of collection and regular-use medications were recorded. Almost half (47.4%) of chronic patients reported using immunosuppressive or anti-inflammatory medication, while only 15.4% of individuals in the acute group reported use of these types of medication. The most commonly reported medications in the chronic group included opioids, corticosteroids, and antidepressants (Appendix A). Additionally, some patients reported using muscle relaxants as complementary therapy.

Complementary therapies such as acupuncture, physical activity, and physiotherapy, aimed at improving joint mobility and recovery, were also reported in the chronic group (Appendix A).

### 3.3. Clinical Data Suggests Possible Neuropathic Disorder in CHIKV Chronic

Pre-existing comorbidities and additional manifestations were also reported for both groups and are detailed in Figure 1C. Out of 19 chronic patients, 10 (50.3%) presented with high blood pressure (SAH). Obesity, type 2 diabetes mellitus, and heart disease were reported by 10.6% of patients. Interestingly, one patient (5.3%) reported a diagnosis of dementia. Concerning acute patients, the frequency of comorbidities was lower, with only two (15.4%) individuals having hypertension and one (7.52%) having type 1 diabetes, respectively (Figure 1C), probably due to the younger age among acute patients. No additional data on pre-existing diseases were available for 10 (76.9%) acute individuals.

Interestingly, additional manifestations were reported in the chronic group. After acute infection, five chronic patients (26.5%) developed tingling and symptoms compatible with neuropathic disease, such as numbness (10.6%), loss of strength (10.4%), and cramps (5.3%) (Figure 1D). Due to intense and constant pain, three patients (15.9%) developed depression, while 15.9% reported a diagnosis of postchikungunya arthritis and joint deformities, mainly affecting fingers and toes (Figure 1D).

### 3.4. Expression Profile and Coexpression Analysis Identified Six Different Gene Sets

We used whole-blood RNA-seq to globally characterize the transcriptional changes occurring during the acute and chronic phases of chikungunya virus infection, focusing on the expression of proinflammatory mediators, such as cytokines and chemokines, as well as differential regulatory and immunological mechanisms correlated with different clinical stages. Detailed statistics of mapping and gene counts are available in Appendix A.

When comparing the differential expression between chronic and acute patients against the control population, we found 2675 were found as differentially expressed (DE) (Appendix A). For individuals with chronic disease, 2678 were differentially expressed, with 1727 (64.48%) showing elevated expression compared to controls (Appendix A). In the acute group evaluation, 233 targets exhibited a positive LogFoldChange, indicating overexpression, while 130 (0.054%) showed reduced expression (Appendix A and Appendix A). Interestingly, when comparing the differentially expressed transcripts between individuals in acute and chronic phases, 2676 genes were found to be either over- or underexpressed (Appendix A and Appendix A). Among these DE genes, 1725 genes (64.46%) were more abundant in the acute phase, while 951 (35.54%) were less expressed.

Although the difference in gene expression found between the groups was significant (*p*-value < 0.005), it was slightly associated with a small variation in the read count (Appendix A). In light of this, we carried out a more comprehensive evaluation seeking to describe the coexpression gene modules. Our aim was to understand better the relationship between these genes and the cellular and molecular processes they are involved in.

Using the Weighted Gene Coexpression Network Analysis (WGCNA) (version 1.72-5) [49], we identified 31 gene modules (M1–M31) that were differentially correlated with the Chikungunya infection stages investigated in the study (*p*-values < 0.005) (Figure 2A,B). Based on the *p*-values, six main modules were selected, namely M1, M17, M21-22, M28, and M30 (Figure 2B).

Modules M1, M17, and M30 were positively related to individuals in the acute phase and negatively associated with chronic patients or the control group, respectively (Figure 2B). Additionally, three modules suggest a greater correlation with individuals chronically impacted by the Chikungunya virus: modules 21, 22, and 28, with M21 and M28 negatively related to individuals in the acute phase. For both acute and chronic, the module sizes (higher than 500 and up to 1000 genes) (Figure 2A) suggest the complexity of disease stage scenarios.

### 3.5. CHIKV Acute Individuals Appear to Show Increase in Mitochondrial Respiratory Chain Genes, Besides Antiviral Expression Profile

The acute phase of Chikungunya virus infection is characterized by a considerable increase in proinflammatory cytokines and mediators responsible for initiating the antiviral response. However, CHIKV is also capable of promoting changes in basic energetic and transcriptional regulation processes, aiming to enhance virus replication and dissemination. In this study, we identified complex gene modules associated with hypermethylation, mitochondrial genes related to the electron transport chain and oxidative phosphorylation, as well as clusters associated with endosomes and methylosomes and splicing (Appendix A).

The M30 module consists of 2135 genes, of which 213 formed gene clusters (clusters 1–20) (Figure 2C). This module predominantly features genes involved in mitochondrial processes such as NADH to ubiquinone electron transport, proton-driven ATP synthesis, the aerobic electron transport chain, and oxidative phosphorylation (Appendix A, Appendix A). Key genes like COX5A, COA3, and CYCS contribute to the cytochrome c oxidase complex, driving oxidative phosphorylation, while TXN2 regulates mitochondrial reactive oxygen species, apoptosis, and cell viability (Appendix A). The M30 module also points to apoptosis regulation with genes like AIFM1, EIF5A, and LGALS1. AIFM1 may regulate caspase-independent apoptosis by relocating from the mitochondria to the nucleus, while EIF5A is linked to both p53/TP53-dependent and TNF-α-mediated apoptosis [53]. LGALS1 is a known regulator of T-cell apoptosis, cell proliferation, and differentiation [54,55]. However, despite the presence of apoptosis-related genes, enrichment analysis did not detect apoptosis pathways (Appendix A).

The M17 module (Figure 2D) was found to be enriched in proteins and mediators associated with the response to interferons (Appendix A). Enrichment analysis revealed increased biological processes such as the negative regulation of IP-1 production (2 of 3 genes, strength 2.89; FDR = 0.0024) and the negative regulation of chemokine (C-X-C motif) ligand 2 production (2 of 4 genes, strength 2.76; FDR = 0.0034). In acute phase individuals, we observed increases in antiviral pathways such as the OAS antiviral response (3 of 9 genes, strength 2.59; FDR ≤ 0.0001), interferon α/β signaling (11 of 71 genes, strength 2.25; FDR ≤ 0.0001), positive regulation of monocyte chemotactic protein-1 production (2 of 14 genes, strength 2.22; FDR = 0.02), negative regulation of viral genome replication (9 of 56 genes, strength 2.27; FDR ≤ 0.0001), and antiviral innate immune response pathway (3 of 22 genes, strength 2.2; FDR = 0.0005). Moreover, the gene EPSTI1 (epithelial-stromal interaction protein 1), important for M1 macrophage polarization, was also identified [40].

Of the 16 genes in this module, 13 perform antiviral functions mediated or induced by interferons, including the OAS family, which bind PPP-RNA to inhibit viral mRNA expression [56,57,58]. OAS2 and OAS3, activated by sRNA, also degrade cellular and viral RNA via ribonuclease L activation, halting viral replication. OAS3 may also influence apoptosis and gene regulation. An enriched pathway in our dataset is 7-methylguanosine cap hypermethylation (6 of 8 genes, strength: 1.85; FDR ≤ 0.0001) [56,57,58]. The module also includes IFIT1-3 and IFIT-5 genes, which complement OAS1-3 by inhibiting viral mRNA and inducing type I interferons and proinflammatory cytokines [59,60].

### 3.6. CHIKV Chronic Individuals Seem to Show Differentiation in Histones Methylation and Th-17 Regulation Genes

The factors that trigger the persistence of symptoms postinfection by Chikungunya virus in approximately 50% of the infected population and which can last months or years after the resolution of the acute phase remain unclear. In this study, we identified three overexpressed complex gene modules associated with chronic status: modules M21, M22, and M28.

Module M21, linked to erythrocyte development, showed a positive correlation with chronically affected CHIKV individuals and a negative correlation in acute-phase patients (Figure 2B). Pathways associated with erythrocyte development may reflect differences in cellularity (lymphocyte/erythrocyte ratio) between these phases. Notably, genes like RNF10 and TNS1 were identified (Appendix A). RNF10, involved in MAG expression, may influence Schwann cell differentiation and myelination, potentially aiding nervous tissue repair in chronic individuals, consistent with neuropathic pain observed in acute patients. TNS1 may regulate cell migration and cartilage development, linking signal transduction pathways to the cytoskeleton [61,62].

Module M22 displayed enrichment in biological processes related to histone methylation, affecting transcriptional regulation (Appendix A, Appendix A). Of 46 genes in this module, eight influence methylation on lysine residues, while four are involved in epigenetic marks. KMT2C and KMT2D methylate Lys-4 of histone H3, acting as nuclear receptor coactivators and marking for transcription [63,64]. KDM3B and TET2 remove methyl groups, with roles in active DNA demethylation [65]. Studies have shown viral infections, such as SARS-CoV-2 and arboviruses, can alter epigenetic profiles even after resolution [66,67].

In the M28 module (85 genes), pathways related to immune responses, including interleukin signaling (IL-2, 4, 6, 7, 9, 13, 15, 21) and CD46 receptor, were identified (Figure 2E, Appendix A, Appendix A). Notably, three genes are linked to the regulation of the Th-17 immune profile (FDR = 0.018) and T-helper 17 cell lineage commitment (FDR = 0.0027), including IL-17RA, IL-6R, and STAT3, which regulate immune responses by inducing inflammatory chemokines and cytokines [56,57,68]. Chronic CHIKV patients also show upregulation of genes associated with neuronal death (FOXO3, MCL1, ZNF746, and PICALM) and T-cell differentiation, which may contribute to secondary neuropathy symptoms [69,70].

## 4. Discussion

Chikungunya fever, a disease caused by CHIKV infection, is characterized by an acute phase marked mainly by a high fever (>38.9 °C) of sudden onset. In this phase, classic symptoms of the infection, including arthralgia, become prominent and debilitating [71,72]. Additionally, as CHIKV viremia increases, the inflammatory condition becomes exacerbated, aiming to combat and restrict viral spread. Innate immunity cells are primarily responsible for attempting to suppress replication before an acquired immune response is triggered [73,74,75,76]. In this study, we aimed to identify differentially expressed transcripts in the peripheral blood of patients during distinct CHIKV infection phases.

The symptoms reported by our patients with acute infection align with those documented in the literature, especially when compared to those described in individuals infected by the CHIKV-ECSA genotype, the most prevalent genotype circulating in Brazil and in the municipality of Feira de Santana [14,15,16,17,18,19,20,21,22,23]. However, the persistence of post-infection joint symptoms, as well as the appearance of additional manifestations, lead to a daily pain sensation, resulting in the onset of depression in chronic individuals. Interestingly, a significant number of patients reported the use of amitriptyline and pregabalin (Appendix A), which are among the first-line recommended drugs for neuropathic pain in addition to being used for depressive disorders [77,78].

We found that the spectrum of rheumatic and musculoskeletal disorders associated with post-chikungunya can include tunnel syndromes, modification of the joint axes, compromise of movement, and sometimes deformities [79,80,81,82]. This scenario has been reported in previous outbreaks in Réunion and India, indicating that the persistence of symptoms in a chronic postchikungunya stage impairs the individual’s quality of life [82,83,84]. Patients with postchikungunya rheumatoid arthritis may experience clinical improvement with methotrexate and hydroxychloroquine, which are used to treat classic rheumatoid arthritis [85]. Although methotrexate has been widely used in the clinic, it has not been reported in our population (Appendix A).

The acute phase is marked by the overexpression of inflammatory mediators, such as IFN-α and γ, IL-2R, IL-6, and IL-9, as well as the presence of IL-8, IP-10, MCP-1, MIG, CXCL-9, CXCL10, MIP-1α and β chemokines, and growth factors G-CSF, GM-CSF, HGF, and VEGF-A [40,86,87,88,89,90,91]. In a recent study carried out in Brazil, acute patients have shown a similar expression profile, as reported here [92], for the same Chikungunya virus lineage (East-Central South African—ECSA). Additionally, due to the lack of information regarding the cycle threshold of CHIKV genome amplification, it was not possible to correlate the CHIKV viral load, the presence of multiple sites of arthralgia and myalgia, and the immune profiling.

After establishing an effective cellular infection, viral replication is initially limited by a rapid and robust production of proinflammatory mediators, such as Interferon-β and the downstream of signaling molecules. Despite rapid immune response against the virus triggered by interferons in permissive cells, CHIKV can effectively evade the cellular control mechanisms. One of the cytopathic effects caused by CHIKV is apoptosis, evidenced by the presence of numerous active caspase-3 and CHIKV double-positive HeLa cells [93]. Although the precise events leading to apoptosis are not yet fully understood, Jaffar-Bandjee et al. (2009) [93] suggest that both the intrinsic (mitochondrial) and extrinsic (Fas/TRAIL-like) apoptosis pathways are involved following CHIKV infection. Studies estimate that within 24 h postinfection, fibroblasts and stromal cells undergo apoptosis [94,95]. The intrinsic pathway, mediated by the release of cytochrome c in the mitochondria, activates caspase-9 and caspase-3 subsequentially; the extrinsic pathway seems to contribute, given the several upregulated death receptors and ligands. The increase in expression related to mitochondrial respiratory chain genes may be also related to the activation of the apoptosis intrinsic pathway found in the acute cases. The virus utilizes this cellular response to increase its rate of infection, and since the viral particles are sequestered within the apoptotic blebs, they escape recognition by the immune system. The engulfment of the apoptotic blebs by phagocytic cells, such as macrophages, can enhance infection silently [96]. Moreover, this mechanism triggers a heightened and dysregulated adaptive immune response, which may result in autoimmunity targeting self-antigens contained within apoptotic bodies [97,98], a process that requires further explanation as a potential trigger for an autoimmune response. Furthermore, the metabolic disturbances observed in these genes may be also associated with the exacerbated use of energy resources by host cells during cell infection [99,100,101,102].

In acute CHIKV patients, we observed an increase in the cap hypermethylation pathway, which involves the addition of the m7GMP cap structure (Cap0) to RNA. This structure is crucial for mRNA stability, processing, and translation [103,104,105] and is essential for viral protein synthesis. Additionally, capping helps viruses evade host immunity by preventing the recognition of terminal RNA phosphates by RIG-I and IFIT1 receptors [60,104,105,106]. In alphaviruses, the nsP1 protein adds cap0 structures to viral RNAs [106,107,108], but not all viral RNAs are capped, suggesting uncapped RNAs may modulate the immune response [109,110]. The increased expression of IFIT and RIG-1-like receptor genes (Appendix A) and those involved in cap hypermethylation highlights an antiviral response aimed at controlling replication and viral protein translation. Further research on RNA capping in CHIKV infections is needed to develop treatments targeting this mechanism [104]. Overall, transcriptional changes during the acute phase rapidly engage immune and metabolic pathways to combat the infection.

Chronic joint involvement in Chikungunya virus infection shows similarities to rheumatoid arthritis, including shared symptoms and peripheral blood mononuclear cell profiles. Some studies attribute persistent symptoms to high levels of IL-6, GM-CSF, INF-α, and IL-17, a finding supported by our data. Other hypotheses suggest that viral RNA in the synovium and low-level replication in reservoir cells, like muscle satellite cells and synovial macrophages, drive chronic inflammation and autoimmunity [89,111,112]. More recently, epigenetic changes, such as DNA hypomethylation and histone modifications, have been described in rheumatoid arthritis and other viral infections like SARS-CoV-2 [66], though these mechanisms are less explored in Chikungunya [61].

Our findings also suggest that T-cell differentiation during the transition from acute to chronic infection may play a role in sequelae. Acute infection often involves lymphopenia and a predominance of CD8+ T cells expressing CD69, CD107a, granzyme B, and perforin, targeting virus-infected cells [36,37,113,114]. However, excessive antigenic exposure in the acute phase can exhaust these lymphocytes, reducing their efficacy in the chronic phase and potentially contributing to persistent symptoms [114,115]. Moreover, excessive cellular infiltrates, particularly from activated memory T-cells, can expose articular tissues to cytokines and chemokines like CXCL-9, CXCL-10, CCL2, IL-6, and IL-10, leading to tissue injury and edema [115,116,117]. While the role of T-cells in chronic Chikungunya infection remains unclear, an animal model study indicated that the absence of CD4+ T-cells reduces joint pathology [39]. Additionally, key lymphocyte subsets, such as Th17 and regulatory T cells (Tregs), may be involved in chronic infection. Th17 cells promote inflammation by producing IL-17, IL-22, and IL-23, while Tregs counterbalance by releasing anti-inflammatory cytokines like IL-10 and TGF-β, easing joint symptoms. Cytokine imbalance, particularly elevated IL-1b, IL-6, and IL-17, appears to drive persistent disease, as shown in multiple studies [37,40,118,119]. However, a study in Thailand during the 2009–2010 outbreak did not find a similar increase in IL-17, which could be due to methodological differences or a different CHIKV strain [120].

Even though our study highlights important candidates’ genes in the regulation and prognosis markers from different CHIKV infection phases, the small sample size may have reduced our ability to correctly identify these markers [121]. Furthermore, factors associated with the virus, such as strain and viral load during the acute phase, as well as factors associated with the sample type [122] and the study population [122], may add additional bias to the analyses. Above all, the use of immunosuppressive medication by the chronic population may affect gene expression and module identification [123], acting as a potential confounding factor for reproducibility and specificity of gene set analysis.

Taken together, our results highlight the high complexity of the distinct disease trajectories caused by the Chikungunya virus. They demonstrate the regulation of new genetic modules that can expand our understanding of the acute and chronic stages. However, further longitudinal studies on the role of immune regulation and different regulatory mechanisms are pivotal in order to provide a better understanding of the disease’s progression and to improve clinical management and patients’ quality of life.

## Figures and Tables

**Figure 1 genes-15-01365-f001:**
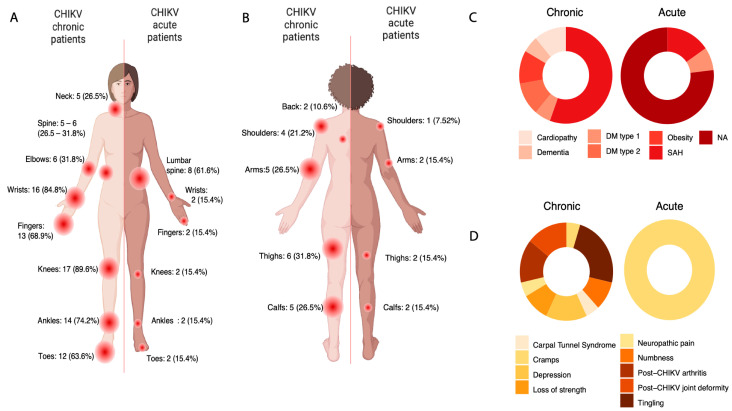
Symptoms and comorbidities panel reported by patients manifesting chronic and acute signs of Chikungunya virus infection. (**A**) Compromised joints in individuals with chronic and acute Chikungunya virus infection. For 10 acute individuals, it was not possible to retrieve data regarding the arthralgia sites. (**B**) Points of myalgia reported by chronic and acute individuals for Chikungunya virus infection. For 10 acute individuals, it was not possible to retrieve data about the myalgia points. (**C**) Pre-existing comorbidities described by individuals chronically affected by Chikungunya virus and individuals in the acute phase; (**D**) Secondary manifestations indicated by chronic and acute individuals for Chikungunya virus reported by clinical examination and medical record.

**Figure 2 genes-15-01365-f002:**
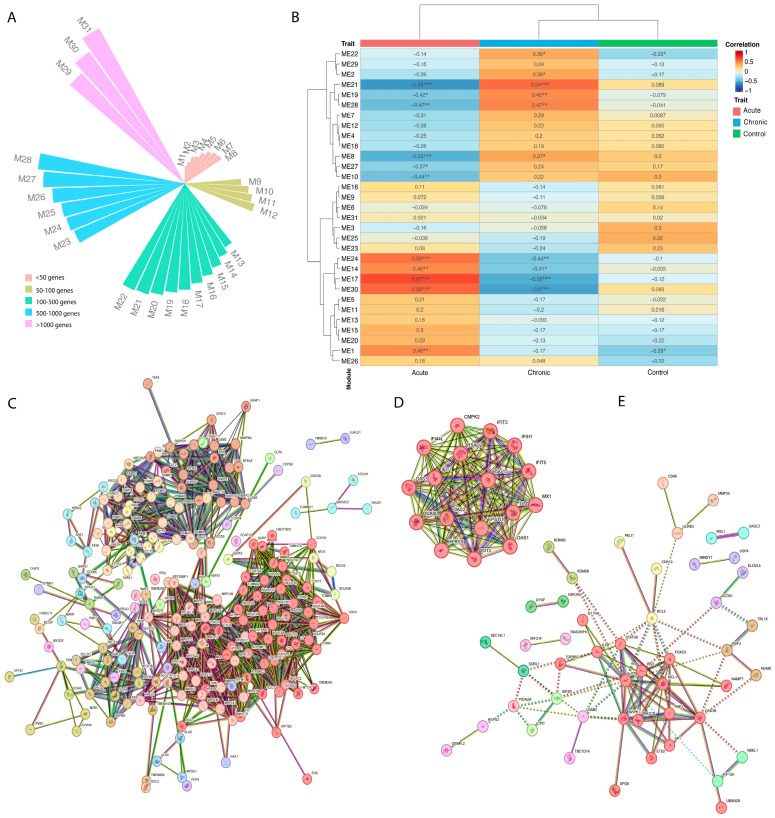
Weighted Gene Coexpression Network Analysis (WGCNA) for distinct Chikungunya virus infection stages and control group. (**A**) Composition of gene modules found differentially coexpressed in acute and chronic individuals for Chikungunya virus. In red are modules with a size of less than 50 genes; in olive modules comprising 50–100 genes; in green modules containing up to 500 genes; in blue gene size modules between 500 and 1000; and, in magenta, modules larger than 1000 genes. (**B**) Coexpression trait heatmap of the 31 modules was found to differentially coexpress in patients in the acute stage of infection and individuals in the chronic stage. The rows represent each of the 31 modules found, while the columns represent the different clinical stages of Chikungunya virus infection. Positive correlations are indicated in shades of red, while negative correlations are in shades of blue. *: *p*-values < 0.05; **: *p*-values < 0.01; ***: *p*-values < 0.001. (**C**) Coexpression module M30 positively correlated with CHIKV acute patients and negatively correlated with chronic individuals. Image obtained using the STRING software. Genes are colored based on subclustering identification using the MCL clustering algorithm (Appendix A). (**D**) Coexpression module M17 positively correlated with CHIKV acute patients and negatively correlated with chronic individuals. Image obtained using the STRING software. Genes are colored based on subclustering identification using the MCL clustering algorithm (Appendix A). (**E**) Coexpression module M28 negatively correlated with acute patients and positively correlated with CHIKV chronic individuals. Image obtained using the STRING software. Genes are colored based on subclustering identification using the MCL clustering algorithm (Appendix A).

**Table 1 genes-15-01365-t001:** Epidemiological and clinical description of chronic, acute, and control individuals for CHIKV infection.

Patients (n = 37)			
	Chronic (n = 19)	Acute (n = 13)	Control (n = 5)
Epidemiological			
Mean age (SD)	60.4 (±13.74)	44.0 (±20.35)	57.8(±20.57)
Gender			
Female	17 (89.6%)	7 (54%)	1 (20%)
Male	2 (10.4%)	6 (46%)	4 (80%)
NA	-	-	-
Onset of symptoms (SD)	-	5 days	-
Schooling			
NA	1 (5.3%)	11(84.1%)	0 (0%)
Incomplete Elementary School	2 (10.6%)	0 (0%)	0 (0%)
Complete Elementary School	2 (10.6%)	0 (0%)	0 (0%)
Incomplete High School	5 (26.3%)	0 (0%)	1 (20%)
Complete High School	5 (26.3%)	2 (15.9%)	2 (40%)
Incomplete Higher Education	2 (10.6%)	0 (0%)	0 (0%)
Complete Higher Education	2 (10.6%)	0 (0%)	1 (20%)
Postgraduate studies	0 (0%)	0 (0%)	1 (20%)
Household income			
Less than minimum wage	0 (0%)	1 (7.52%)	0 (0%)
From 1 to 2x minimum wages	18 (94.8%)	2 (15.4%)	5 (100%)
From 3 to 4× minimum wages	1 (5.3%)	0 (0%)	0 (0%)
More than 4× minimum wages	0 (0%)	0 (0%)	0 (0%)
NA	0 (0%)	10 (77.08%)	0 (0%)
Use of Immunosuppressive or Anti-inflammatory Medication			
Yes	9 (47.4%)	2 (15.4%)	0 (0%)

## Data Availability

The original data presented in the study are openly available in the European Nucleotide Archive (ENA), submitted under accession code PRJEB74939. Further inquiries can be directed to the corresponding authors.

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
