# Peer review of "Unraveling the Complexity of Chikungunya Virus Infection Immunological and Genetic Insights in Acute and Chronic Patients"

_genes, 2024, doi:10.3390/genes15111365_

Round 1

Reviewer 1 Report

Comments and Suggestions for Authors

Dear authors.

Congratulation for the manuscript, I enjoy reading it.

The infection with chikungunya virus (CHIKV) has caused  a significant number of infections worldwide. Your manuscript analyses the emergence of the CHIKV-ECSA genotype in Brasil.

Your study aimed to shed new light on the host immune response by examining the whole blood transcriptomic profile of both CHIKV-acute and chronically infected individuals from Brazil.

You  revealed a complex symptomatology characterized by arthralgia and post-chikungunya neuropathy in individuals with chronic sequelae, particularly affecting women.

Analysis of gene modules suggests heightened metabolic processes  in patients with acute disease and in individuals with chronic manifestations  a distinct pattern of histone methylation and T-cells differentiation are seen.

Overall, our findings underscore the intricate nature of disease progression in CHIKV infections, highlighting the diverse scenarios encountered in both acute and chronic stages.

Moreover, our identification of specific gene modules linked with viral disease offers valuable insights into the underlying mechanisms driving these distinct disease manifestations.  

The manuscript is well written, study design and material and methods are well presented.

Discussion and conclusion support the results.

Figures are presented clearly.

All references are appropriate.

Author Response

Response to Reviewers

Dear Editor,

We are pleased that the editorial assessment of our manuscript was largely favourable and thank the reviewers and editors for the constructive comments. We have addressed all comments (see below) and we believe that the resulting manuscript is much improved. We hope you will consider the revision to be suitable for publication in Genes Journal.

Reviewer #1 (Comments for the Author):

Dear authors.

Congratulation for the manuscript, I enjoy reading it.

The infection with chikungunya virus (CHIKV) has caused  a significant number of infections worldwide. Your manuscript analyses the emergence of the CHIKV-ECSA genotype in Brasil.

Your study aimed to shed new light on the host immune response by examining the whole blood transcriptomic profile of both CHIKV-acute and chronically infected individuals from Brazil.

You  revealed a complex symptomatology characterized by arthralgia and post-chikungunya neuropathy in individuals with chronic sequelae, particularly affecting women.

Analysis of gene modules suggests heightened metabolic processes  in patients with acute disease and in individuals with chronic manifestations  a distinct pattern of histone methylation and T-cells differentiation are seen.

Overall, our findings underscore the intricate nature of disease progression in CHIKV infections, highlighting the diverse scenarios encountered in both acute and chronic stages.

Moreover, our identification of specific gene modules linked with viral disease offers valuable insights into the underlying mechanisms driving these distinct disease manifestations.  

Comment 1: The manuscript is well written; study design and material and methods are well presented.

Reply: We thank the reviewer for the positive comments.

Comment 2: Discussion and conclusion support the results.

Reply: We thank the reviewer for the positive comments.

Comment 3: Figures are presented clearly.

Reply: We thank the reviewer for the positive comments.

Comment 4: All references are appropriate.

Reply: We thank the reviewer for the positive comments.

Reviewer 2 Report

Comments and Suggestions for Authors
  1. Clarity and Structure of the Abstract:
    The abstract contains significant information, but it lacks focus. It would be beneficial to streamline it by emphasizing the main findings and their implications in a more concise manner. Additionally, some sentences are dense and could be simplified for easier comprehension by a broader audience.
  2. Introduction:
    • The introduction provides a comprehensive background on Chikungunya virus (CHIKV). However, it could benefit from a more structured flow. The transition from general information about the virus to specific details about the study is somewhat abrupt. Adding a clearer statement of the research objective towards the end would help set the stage for the methods and results.
    • The description of the different CHIKV lineages is useful, but the relevance of this information to the study needs to be more explicit.
  3. Methods:
    • The methods section is generally well-detailed, but some areas could use clarification. For example, the criteria for selecting patients for the acute and chronic phases are not fully explained. Further elaboration on the inclusion and exclusion criteria for participant selection would be beneficial.
    • The data analysis portion (especially for differential gene expression and coexpression network analysis) could be further elaborated for clarity. Some of the statistical choices, such as the threshold for significance, should be justified more clearly.
  4. Results:
    • The results are robust, but some sections are quite dense. For example, the gene coexpression network analysis involves multiple gene modules, which may overwhelm the reader. Presenting the results in a more narrative style, focusing on key genes and pathways, would make the findings more digestible.
    • There is a repetition of certain information in the results and discussion sections. Streamlining the presentation of results, while referring to supplementary figures and tables, could enhance readability.
    • It would be beneficial to provide more interpretation of the significance of the findings related to mitochondrial respiratory chain genes and apoptosis. The implications for the acute and chronic phases of infection should be more explicitly linked to the broader disease mechanisms.
  5. Discussion:
    • The discussion provides an in-depth analysis of the results, but it sometimes feels disconnected from the rest of the paper. The narrative could be made more cohesive by linking it more clearly with the introduction and the stated research questions.
    • While the paper provides important findings, the practical applications of these findings could be highlighted more. For instance, how can this knowledge inform clinical practice or future research in the field of CHIKV or other similar viruses?
    • The limitations of the study, including the relatively small sample size and the potential confounding factors (e.g., medication use, genetic factors), should be discussed more thoroughly.

Author Response

Response to Reviewers

Dear Editor,

We are pleased that the editorial assessment of our manuscript was largely favourable and thank the reviewers and editors for the constructive comments. We have addressed all comments (see below) and we believe that the resulting manuscript is much improved. We hope you will consider the revision to be suitable for publication in Genes Journal.

Reviewer #2 (Comments for the Author):

Clarity and Structure of the Abstract:

Comment 1: The abstract contains significant information, but it lacks focus. It would be beneficial to streamline it by emphasizing the main findings and their implications in a more concise manner. Additionally, some sentences are dense and could be simplified for easier comprehension by a broader audience.

Reply: We thank the reviewer for comment. The abstract was reviewed and improvements were made in order to simplified the comprehension.

Lines 56 to 65: “Analysis of gene modules suggests heightened metabolic processes, represented by an increase of NADH, COX5A, COA3, CYC1, and cap methylation in patients with acute disease. In contrast, individuals with chronic manifestations exhibit a distinct pattern of histone methylation, probably mediated by NCOA3 in coactivation of different nuclear receptors, KMT2 genes, KDM3B and TET2, and with alterations in the immunological response, major leaded by IL-17RA, IL-6R and STAT3 Th17 genes. Our results emphasize the complexity of CHIKV disease progression, demonstrating the heterogeneous gene expression and symptomatologic scenario across both acute and chronic phases. Moreover, the identification of specific gene modules associated with viral pathogenesis provides critical insights into the molecular mechanisms underlying these distinct clinical manifestations.”

Introduction:

Comment 2: The introduction provides a comprehensive background on Chikungunya virus (CHIKV). However, it could benefit from a more structured flow. The transition from general information about the virus to specific details about the study is somewhat abrupt. Adding a clearer statement of the research objective towards the end would help set the stage for the methods and results.

Reply: We thank the reviewer for comment. Improvements were done in the introduction.

Lines 90 to 93: “The vast majority of Chikungunya infected patients may experience an asymptomatic infection (24). However, for those who shows clinical manifestations, the acute phase may be marked by a sudden onset of fever, followed by cutaneous manifestations (rash), fatigue, and debilitating polyarthralgia.”

Lines 101 to 103: “Even though several studies have focused on characterizing the immune response to CHIKV infection, describing the immunobiological mechanisms, the natural history of the disease and factors associated with chronicity remains poorly understood (2,30–33).”

Lines 113 to 114: “To further elucidate virus-host mechanisms, transcriptomic approaches offer a promising means of accessing molecular and immunological profiles in disease cases.”

Lines 119 to 125: “In this study, we aimed to provide new insights of the molecular mechanisms and possible genes modules involved in different phases of Chikungunya infection. For that, we used transcriptomics to characterize acute and chronic CHIKV infection in individuals from Feira de Santana, Bahia, Brazil—the municipality where ECSA-CHIKV was introduced into the country and served as a source of transmission to several regions heavily impacted by the co-circulation of Dengue and Zika viruses.”

Comment 3: The description of the different CHIKV lineages is useful, but the relevance of this information to the study needs to be more explicit.

Reply: We thank the reviewer for comment. Improvements were done in the introduction in order to clarify the importance of CHIKV lineage in disease progression.

Lines 77 to 89: “It can be classified into four distinct lineages or genotypes based on genomic differences: (i) the West African lineage; (ii) the East/Central/South African (ECSA) lineage; (iii) the Asian lineage, and (iv) the Indian Ocean lineage (IOL) (8–10). Although there is no consensus in the literature regarding the impact of the viral lineage on the severity and prognosis of the disease, CHIKV-ECSA infections seem to be associated with more severe symptoms compared to other lineages (11). Since its initial cases, reported in Tanzania in 1953, CHIKV infections have been reported worldwide (3,9,12,13), with almost 4 million probable cases in the Americas alone from 2013 to May 2024 (3,13). In Brazil, the most affected country in the continent (3,13), the local introduction of the CHIKV-ECSA genotype was first detected in the municipality of Feira de Santana, Northeast region in 2014. Since then, the ECSA genotype has been reported in several states across the country, posing a serious threat to public health (14–23), given the uncertain impact of CHIKV-ECSA infection in more severe cases (11).”

Methods:

Comment 4: The methods section is generally well-detailed, but some areas could use clarification. For example, the criteria for selecting patients for the acute and chronic phases are not fully explained. Further elaboration on the inclusion and exclusion criteria for participant selection would be beneficial.

Reply: We thank the reviewer for comment. The authors attempted to recruit homogeneous groups of acute, chronic, and control individuals in terms of gender and age. However, given the unavailability of acute individuals at the time of recruitment, mainly due to the increase in Dengue cases in the Feira de Santana region, masking Chikungunya cases, it was not possible to achieve adequate homogenization. In addition, the enrollment of control individuals was intensely impacted by the number of affected women within the families, most of whom suffered from post-CHIKV sequelae.

In order to clarify the enrollment criteria, the section has been rewritten.

Lines 143 to 158: “Acute cases suspected of Chikungunya infection underwent nucleic acid extraction and purification using the Reliaprep Viral TNA kit (Promega) and subsequent laboratory confirmation through multiplex RT-qPCR assay targeting Zika, Dengue, and Chikungunya viruses (ZDC molecular kit, Bio-Manguinhos). Individuals who tested positive for Chikungunya virus up to 5 days after the onset of symptoms were invited to participate. Patients co-infected with other arbovirus pathogens in the arboviruses diagnose panel were excluded from the analysis to mitigate potential biases in immunological response assessment.

As an inclusion criterion for the chronic group, all chronic individuals who demonstrated, by clinical report and physical exam, persistence of symptoms for one or more years following CHIKV infection were enrolled. Additionally, chronical individuals should had been under medical supervision since the convalescent phase (approximately 90 days post-symptom onset) in order to retrieve the medical report over years. To control for genetic and behavioral biases, healthy individuals genetically related to chronic cases (members of the same family), who had a previous history of CHIKV infection but did not develop chronic disease, were selected as a control group.”

Comment 5: The data analysis portion (especially for differential gene expression and coexpression network analysis) could be further elaborated for clarity. Some of the statistical choices, such as the threshold for significance, should be justified more clearly.

Reply: We thank the reviewer for the insightful comment. We acknowledge that further elaboration on the data analysis portion, particularly regarding differential gene expression and coexpression network analysis, would improve the clarity of our manuscript.

Lines 206 to 250: “2.5 Differential Expression Analysis

Before performing the differential expression analysis, the StringTie output was processed using the prepDR.py3 script available on the StringTie website. To identify differentially expressed genes across various stages of Chikungunya infection, we utilized the edgeR package (version 4.0.16) (48) in the R statistical environment. The workflow included normalizing read counts, estimating dispersion parameters, and fitting statistical models to detect genes showing significant changes in expression between conditions.

Dispersion estimation in edgeR (48) was conducted in three stages. First, common dispersion was estimated using the estimateGLMCommonDisp function, which calculates a single dispersion value for all genes, assuming shared variability across samples. Next, trend dispersion was assessed using the estimateGLMTrendedDisp function to capture any relationship between gene abundance and variability. Finally, gene-specific (tagwise) dispersion was estimated using the estimateGLMTagwiseDisp function, providing individual dispersion estimates for each gene.

Statistical models were then fitted using the glmFit function, which applies a negative binomial generalized log-linear model (GLM) to the read count data. To identify significant differences in gene expression between conditions, we used the likelihood ratio test implemented in the glmLRT function. A contrast matrix was constructed to specifically compare gene expression between control versus acute and control versus chronic infection stages.

To ensure the reliability of the results, the false discovery rate (FDR) was controlled using the Benjamini-Hochberg correction. Genes were considered differentially expressed if they showed an FDR of < 0.05 and a log fold-change (logFC) greater than |1|.

2.6 Gene Coexpression and Functional Enrichment Pathway Analysis

Following the differential expression analysis, we utilized the Weighted Gene Coexpression Network Analysis (WGCNA) package (version 1.72-5) (49)  in R to identify gene coexpression modules. Prior to this, the gene expression data were converted to log2 counts per million (log2CPM) to ensure accurate comparisons. We employed the sft_fit function to determine the soft-thresholding power for network construction, utilizing a signed hybrid network type to capture both positive and negative correlations. Pearson correlation was used to calculate the correlation coefficients.

Subsequently, we correlated the identified modules with the various stages of Chikungunya infection using the module_trait_cor function. Modules exhibiting statistically significant correlations (p-value < 0.05) were prioritized for further investigation.

To elucidate the biological functions and pathways associated with the identified coexpression modules, we performed functional enrichment analysis using the gProfiler tool (available at https://biit.cs.ut.ee/gprofiler/, accessed in February 2024) (50,51) and STRING (available at https://string-db.org/, accessed in February 2024). Enriched terms were considered biologically relevant if they had a p-value < 0.05 and a significant number of genes involved in the pathway.

For further exploration of the significantly correlated modules, we applied the Markov Cluster Algorithm (MCL)(52) on STRING, using a graph clustering method based on stochastic flow simulation. An inflation parameter of 3 was used, and the resulting clusters within each module were visualized using the STRING platform. “

Results:

Comment 6: The results are robust, but some sections are quite dense. For example, the gene coexpression network analysis involves multiple gene modules, which may overwhelm the reader. Presenting the results in a more narrative style, focusing on key genes and pathways, would make the findings more digestible.

Reply: We thank the reviewer for the valuable feedback. We acknowledge that some sections, such as the gene coexpression network analysis, may appear dense due to the volume of findings. However, the Results section was structured to reflect the complexity and robustness inherent in describing molecular and cellular mechanisms across distinct phases of viral infection (acute and chronic). The results are divided according to the infection stage, with section titles providing an initial overview of the key topics to be discussed. Specifically, in section 3.4, "Expression profile and coexpression analysis identified six different gene sets," the presentation of multiple gene modules serves as an initial and preliminary overview. These modules were further detailed and discussed in subsequent sections, allowing for a clearer narrative that highlights key genes and possible involved pathways.

In order to facilitate the comprehension, the sections 3.4 Expression profile and coexpression analysis identified six different gene sets; 3.5 CHIKV acute individuals appear to show increase in mitochondrial respiratory chain genes, besides antiviral expression profile; and 3.6 CHIKV chronic individuals seem to show differentiation in histones methylation and Th-17 regulation genes were reduced.

Lines 327 to 361: “We used whole-blood RNA-seq to globally characterize the transcriptional changes occurring during the acute and chronic phases of chikungunya virus infection, focusing on the expression of pro-inflammatory mediators, such as cytokines and chemokines, as well as differential regulatory and immunological mechanisms correlated with different clinical stages. Detailed statistics of mapping and gene counts are available in Supplementary Table 2.

When comparing the differential expression between chronic and acute patients against the control population, we found 2,675 were found as differentially expressed (DE) (Supplementary Figure 1A). For individuals with chronic disease, 2,678 were differentially expressed, with 1,727 (64.48%) showing elevated expression compared to controls (Supplementary Table 3). In the acute group evaluation, 233 targets exhibited a positive LogFoldChange, indicating overexpression, while 130 (0.054%) showed reduced expression (Supplementary Table 4 and Supplementary Figure 1B). Interestingly, when comparing the differentially expressed transcripts between individuals in acute and chronic phases,2,676 genes were found to be either over or under-expressed (Supplementary Figure 1C and Supplementary Table 5). Among these DE genes, 1,725 genes (64.46%) were more abundant in the acute phase, while 951 (35.54%) were less expressed.

Although the difference in gene expression found between the groups was significant (p-value <0.005), it was slightly associated with a small variation in the read count (Supplementary Figure 1). In light of this, we carried out a more comprehensive evaluation seeking to describe the co-expression gene modules. Our aim was to understand better the relationship between these genes, and the cellular and molecular processes they are involved in.

Using the Weighted Gene Coexpression Network Analysis (WGCNA) (version 1.72-5) (49), we identified 31 gene modules (M1 to M31) that were differentially correlated with the Chikungunya infection stages investigated in the study (p-values <0.005) (Figures 2A-B). Based on the p-values, six main modules were selected, namely M1, M17, M21-22, M28 and M30 (Figure 2B).

Modules M1, M17 and M30 were positively related to individuals in the acute phase and negatively associated with chronic patients or the control group, respectively (Figure 2B). Additionally, three modules suggest a greater correlation with individuals chronically impacted by the Chikungunya virus: modules 21, 22 and 28, with M21 and M28 negatively related to individuals in the acute phase. For both acute and chronic, the modules sizes (higher than 500 and up to 1000 genes) (Figure 2A), suggest the complexity of disease stages scenarios.”

Lines 395 to 429: “The M30 module consists of 2,135 genes, of which 213 formed gene clusters (clusters 1–20) (Figure 2C). This module predominantly features genes involved in mitochondrial processes such as NADH to ubiquinone electron transport, proton-driven ATP synthesis, the aerobic electron transport chain, and oxidative phosphorylation (Supplementary Table 9, Supplementary Figure 2). Key genes like COX5A, COA3, and CYCS contribute to the cytochrome c oxidase complex, driving oxidative phosphorylation, while TXN2 regulates mitochondrial reactive oxygen species, apoptosis, and cell viability (Supplementary Table 9). The M30 module also points to apoptosis regulation, with genes like AIFM1, EIF5A, and LGALS1. AIFM1 may regulate caspase-independent apoptosis by relocating from the mitochondria to the nucleus, while EIF5A is linked to both p53/TP53-dependent and TNF-alpha-mediated apoptosis (53). LGALS1 is a known regulator of T-cell apoptosis, cell proliferation, and differentiation (54,55). However, despite the presence of apoptosis-related genes, enrichment analysis did not detect apoptosis pathways (Supplementary Table 9).

The M17 module (Figure 2D) was found enriched in proteins and mediators associated with the response to interferons (Supplementary Table 10). Enrichment analysis revealed increased biological processes such as the negative regulation of IP-1 production (2 of 3 genes, strength 2.89 – FDR = 0.0024) and the negative regulation of chemokine (C-X-C motif) ligand 2 production (2 of 4 genes, strength 2.76 – FDR = 0.0034). In acute phase individuals, we observed increases in antiviral pathways such as the OAS antiviral response (3 of 9 genes, strength 2.59 – FDR = <0.0001), interferon alpha/beta signaling (11 of 71 genes, strength 2.25 – FDR = <0.0001), positive regulation of monocyte chemotactic protein-1 production (2 of 14 genes, strength 2.22 – FDR = 0.02), negative regulation of viral genome replication (9 of 56 genes, strength 2.27 – FDR = <0.0001), and antiviral innate immune response pathway (3 of 22 genes, strength 2.2 – FDR = 0.0005). Moreover, the gene EPSTI1 (Epithelial-stromal interaction protein 1), important for M1 macrophage polarization, was also identified (40).

Of the 16 genes in this module, 13 perform antiviral functions mediated or induced by interferons, including the OAS family, which bind PPP-RNA to inhibit viral mRNA expression (56–58). OAS2 and OAS3, activated by sRNA, also degrade cellular and viral RNA via ribonuclease L activation, halting viral replication. OAS3 may also influence apoptosis and gene regulation. An enriched pathway in our dataset is 7-methylguanosine cap hypermethylation (6 of 8 genes, strength: 1.85 – FDR = <0.0001) (56–58). The module also includes IFIT1-3 and IFIT-5 genes, which complement OAS1-3 by inhibiting viral mRNA and inducing type I interferons and proinflammatory cytokines (59,60).”

Lines 438 to 463: “Module M21, linked to erythrocyte development, showed a positive correlation with chronically affected CHIKV individuals and a negative correlation in acute-phase patients (Figure 2B). Pathways associated with erythrocyte development may reflect differences in cellularity (lymphocyte/erythrocyte ratio) between these phases. Notably, genes like RNF10 and TNS1 were identified (Supplementary Figure 3). RNF10, involved in MAG expression, may influence Schwann cell differentiation and myelination, potentially aiding nervous tissue repair in chronic individuals, consistent with neuropathic pain observed in acute patients. TNS1 may regulate cell migration and cartilage development, linking signal transduction pathways to the cytoskeleton (61,62).

Module M22 displayed enrichment in biological processes related to histone methylation, affecting transcriptional regulation (Supplementary Table 11, Supplementary Figure 4). Of 46 genes in this module, eight influence methylation on lysine residues, while four are involved in epigenetic marks. KMT2C and KMT2D methylate Lys-4 of histone H3, acting as nuclear receptor coactivators and marking for transcription (63,64). KDM3B and TET2 remove methyl groups, with roles in active DNA demethylation (65). Studies have shown viral infections, such as SARS-CoV-2 and arboviruses, can alter epigenetic profiles, even after resolution (66,67) .

In the M28 module (85 genes), pathways related to immune responses, including interleukin signaling (IL-2, 4, 6, 7, 9, 13, 15, 21) and CD46 receptor, were identified (Figure 2E, Supplementary Table 12, Supplementary Figure 4). Notably, three genes are linked to the regulation of the Th-17 immune profile (FDR = 0.018), and T-helper 17 cell lineage commitment (FDR = 0.0027), including IL-17RA, IL-6R, and STAT3, which regulate immune responses by inducing inflammatory chemokines and cytokines (56,57,68). Chronic CHIKV patients also show upregulation of genes associated with neuronal death (FOXO3, MCL1, ZNF746, PICALM) and T-cell differentiation, which may contribute to secondary neuropathy symptoms (69,70).”

Comment 7: There is a repetition of certain information in the results and discussion sections. Streamlining the presentation of results, while referring to supplementary figures and tables, could enhance readability.

Reply: We thank the reviewer for the comment. In response, we have streamlined the presentation of the results to reduce repetition and improve readability. The relevant information has been correctly linked to the corresponding supplementary figures and tables, ensuring a more concise and focused narrative.

Lines 327 to 361: “We used whole-blood RNA-seq to globally characterize the transcriptional changes occurring during the acute and chronic phases of chikungunya virus infection, focusing on the expression of pro-inflammatory mediators, such as cytokines and chemokines, as well as differential regulatory and immunological mechanisms correlated with different clinical stages. Detailed statistics of mapping and gene counts are available in Supplementary Table 2.

When comparing the differential expression between chronic and acute patients against the control population, we found 2,675 were found as differentially expressed (DE) (Supplementary Figure 1A). For individuals with chronic disease, 2,678 were differentially expressed, with 1,727 (64.48%) showing elevated expression compared to controls (Supplementary Table 3). In the acute group evaluation, 233 targets exhibited a positive LogFoldChange, indicating overexpression, while 130 (0.054%) showed reduced expression (Supplementary Table 4 and Supplementary Figure 1B). Interestingly, when comparing the differentially expressed transcripts between individuals in acute and chronic phases,2,676 genes were found to be either over or under-expressed (Supplementary Figure 1C and Supplementary Table 5). Among these DE genes, 1,725 genes (64.46%) were more abundant in the acute phase, while 951 (35.54%) were less expressed.

Although the difference in gene expression found between the groups was significant (p-value <0.005), it was slightly associated with a small variation in the read count (Supplementary Figure 1). In light of this, we carried out a more comprehensive evaluation seeking to describe the co-expression gene modules. Our aim was to understand better the relationship between these genes, and the cellular and molecular processes they are involved in.

Using the Weighted Gene Coexpression Network Analysis (WGCNA) (version 1.72-5) (49), we identified 31 gene modules (M1 to M31) that were differentially correlated with the Chikungunya infection stages investigated in the study (p-values <0.005) (Figures 2A-B). Based on the p-values, six main modules were selected, namely M1, M17, M21-22, M28 and M30 (Figure 2B).

Modules M1, M17 and M30 were positively related to individuals in the acute phase and negatively associated with chronic patients or the control group, respectively (Figure 2B). Additionally, three modules suggest a greater correlation with individuals chronically impacted by the Chikungunya virus: modules 21, 22 and 28, with M21 and M28 negatively related to individuals in the acute phase. For both acute and chronic, the modules sizes (higher than 500 and up to 1000 genes) (Figure 2A), suggest the complexity of disease stages scenarios.”

Lines 395 to 429: “The M30 module consists of 2,135 genes, of which 213 formed gene clusters (clusters 1–20) (Figure 2C). This module predominantly features genes involved in mitochondrial processes such as NADH to ubiquinone electron transport, proton-driven ATP synthesis, the aerobic electron transport chain, and oxidative phosphorylation (Supplementary Table 9, Supplementary Figure 2). Key genes like COX5A, COA3, and CYCS contribute to the cytochrome c oxidase complex, driving oxidative phosphorylation, while TXN2 regulates mitochondrial reactive oxygen species, apoptosis, and cell viability (Supplementary Table 9). The M30 module also points to apoptosis regulation, with genes like AIFM1, EIF5A, and LGALS1. AIFM1 may regulate caspase-independent apoptosis by relocating from the mitochondria to the nucleus, while EIF5A is linked to both p53/TP53-dependent and TNF-alpha-mediated apoptosis (53). LGALS1 is a known regulator of T-cell apoptosis, cell proliferation, and differentiation (54,55). However, despite the presence of apoptosis-related genes, enrichment analysis did not detect apoptosis pathways (Supplementary Table 9).

The M17 module (Figure 2D) was found enriched in proteins and mediators associated with the response to interferons (Supplementary Table 10). Enrichment analysis revealed increased biological processes such as the negative regulation of IP-1 production (2 of 3 genes, strength 2.89 – FDR = 0.0024) and the negative regulation of chemokine (C-X-C motif) ligand 2 production (2 of 4 genes, strength 2.76 – FDR = 0.0034). In acute phase individuals, we observed increases in antiviral pathways such as the OAS antiviral response (3 of 9 genes, strength 2.59 – FDR = <0.0001), interferon alpha/beta signaling (11 of 71 genes, strength 2.25 – FDR = <0.0001), positive regulation of monocyte chemotactic protein-1 production (2 of 14 genes, strength 2.22 – FDR = 0.02), negative regulation of viral genome replication (9 of 56 genes, strength 2.27 – FDR = <0.0001), and antiviral innate immune response pathway (3 of 22 genes, strength 2.2 – FDR = 0.0005). Moreover, the gene EPSTI1 (Epithelial-stromal interaction protein 1), important for M1 macrophage polarization, was also identified (40).

Of the 16 genes in this module, 13 perform antiviral functions mediated or induced by interferons, including the OAS family, which bind PPP-RNA to inhibit viral mRNA expression (56–58). OAS2 and OAS3, activated by sRNA, also degrade cellular and viral RNA via ribonuclease L activation, halting viral replication. OAS3 may also influence apoptosis and gene regulation. An enriched pathway in our dataset is 7-methylguanosine cap hypermethylation (6 of 8 genes, strength: 1.85 – FDR = <0.0001) (56–58). The module also includes IFIT1-3 and IFIT-5 genes, which complement OAS1-3 by inhibiting viral mRNA and inducing type I interferons and proinflammatory cytokines (59,60).”

Lines 438 to 463: “Module M21, linked to erythrocyte development, showed a positive correlation with chronically affected CHIKV individuals and a negative correlation in acute-phase patients (Figure 2B). Pathways associated with erythrocyte development may reflect differences in cellularity (lymphocyte/erythrocyte ratio) between these phases. Notably, genes like RNF10 and TNS1 were identified (Supplementary Figure 3). RNF10, involved in MAG expression, may influence Schwann cell differentiation and myelination, potentially aiding nervous tissue repair in chronic individuals, consistent with neuropathic pain observed in acute patients. TNS1 may regulate cell migration and cartilage development, linking signal transduction pathways to the cytoskeleton (61,62).

Module M22 displayed enrichment in biological processes related to histone methylation, affecting transcriptional regulation (Supplementary Table 11, Supplementary Figure 4). Of 46 genes in this module, eight influence methylation on lysine residues, while four are involved in epigenetic marks. KMT2C and KMT2D methylate Lys-4 of histone H3, acting as nuclear receptor coactivators and marking for transcription (63,64). KDM3B and TET2 remove methyl groups, with roles in active DNA demethylation (65). Studies have shown viral infections, such as SARS-CoV-2 and arboviruses, can alter epigenetic profiles, even after resolution (66,67) .

In the M28 module (85 genes), pathways related to immune responses, including interleukin signaling (IL-2, 4, 6, 7, 9, 13, 15, 21) and CD46 receptor, were identified (Figure 2E, Supplementary Table 12, Supplementary Figure 4). Notably, three genes are linked to the regulation of the Th-17 immune profile (FDR = 0.018), and T-helper 17 cell lineage commitment (FDR = 0.0027), including IL-17RA, IL-6R, and STAT3, which regulate immune responses by inducing inflammatory chemokines and cytokines (56,57,68). Chronic CHIKV patients also show upregulation of genes associated with neuronal death (FOXO3, MCL1, ZNF746, PICALM) and T-cell differentiation, which may contribute to secondary neuropathy symptoms (69,70).”

Comment 8: It would be beneficial to provide more interpretation of the significance of the findings related to mitochondrial respiratory chain genes and apoptosis. The implications for the acute and chronic phases of infection should be more explicitly linked to the broader disease mechanisms.

Reply: We thank the reviewer for comment. In order to provide a better link between mitochondrial respiratory chain genes and apoptosis, we included the paragraph below.

Lines 500 to 523: “After establishing an effective cellular infection, viral replication is initially limited by a rapid and robust production of pro-inflammatory mediators such as Interferon-b and the downstream of signaling molecules. Despite rapid immune response against the virus triggered by interferons in permissive cells, CHIKV can effectively evade the cellular control mechanisms. One of the cytopathic effects caused by CHIKV is apoptosis, evidenced by the presence of numerous active caspase-3 and CHIKV double-positive HeLa cells (24). Although the precise events leading to apoptosis are not yet fully understood, Jaffar-Bandjee et cols (2009)(24) suggest that both the intrinsic (mitochondrial) and extrinsic (Fas/TRAIL-like) apoptosis pathways are involved following CHIKV infection. Studies estimate that within 24 hours post-infection, fibroblasts and stromal cells undergo apoptosis (93,94). The intrinsic pathway, mediated by the release of cytochrome c in the mitochondria, activate caspase-9 and caspase-3, sub-sequentially; while extrinsic pathway seems to contribute given the several up-regulated death receptors and ligands. The increase of expression related to mitochondrial respiratory chain genes may be also related to the activation of the apoptosis intrinsic pathway found in the acute cases. The virus utilizes this cellular response to increase its rate of infection, and since the viral particles are sequestered within the apoptotic blebs, they escape recognition by the immune system. The engulfment of the apoptotic blebs by phagocytic cells, such as macrophages, can enhance infection silently (95).  Moreover, this mechanism triggers a heightened and dysregulated adaptive immune response, which may result in autoimmunity targeting self-antigens contained within apoptotic bodies (96,97), process that requires further explanations as a potential trigger for an autoimmune response. Furthermore the metabolic disturbances observed in these genes may be also associated with the exacerbate use of energy resources of host cells during cell infection (98–101). ”

Discussion:

Comment 9: The discussion provides an in-depth analysis of the results, but it sometimes feels disconnected from the rest of the paper. The narrative could be made more cohesive by linking it more clearly with the introduction and the stated research questions.

Reply: We thank the reviewer for their thoughtful comment. We agree that enhancing the cohesion between the discussion and the earlier sections of the manuscript would strengthen the overall narrative. As this is an exploratory study, our primary goal was to provide additional data on the gene expression profile of individuals infected during the acute phase, as well as those chronically impacted by Chikungunya virus. The intention was to explore and correlate these findings with potential active modules during these different phases of infection.

Given the exploratory nature of the work, the discussion aimed to contextualize our findings within the broader literature, providing hypotheses and directions for future research. However, we understand that making clearer connections between the introduction, research questions, and discussion will help the narrative, emphasizing how our results address the research objectives outlined at the start of the paper. We modified the discussion accordingly to ensure a more cohesive flow throughout the manuscript.

Lines 482 to 490: We found that the spectrum of rheumatic and musculoskeletal disorders associated with post-chikungunya can include tunnel syndromes, modification of the joint axes, compromise of movement and sometimes deformities (79–82). This scenario has been reported in previous outbreaks in Réunion and India, indicating that the persistence of symptoms in a chronic post-chikungunya stage impairs the individual’s quality of life (82–84). Patients with post-chikungunya rheumatoid arthritis may experience clinical improvement with methotrexate and hydroxychloroquine, which are used to treat classic rheumatoid arthritis (85). Although methotrexate has been widely used in the clinic, it has not been reported in our population (Supplementary Table 1).

Lines 500 to 563:“After establishing an effective cellular infection, viral replication is initially limited by a rapid and robust production of pro-inflammatory mediators such as Interferon-b and the downstream of signaling molecules. Despite rapid immune response against the virus triggered by interferons in permissive cells, CHIKV can effectively evade the cellular control mechanisms. One of the cytopathic effects caused by CHIKV is apoptosis, evidenced by the presence of numerous active caspase-3 and CHIKV double-positive HeLa cells (24). Although the precise events leading to apoptosis are not yet fully understood, Jaffar-Bandjee et cols (2009)(24) suggest that both the intrinsic (mitochondrial) and extrinsic (Fas/TRAIL-like) apoptosis pathways are involved following CHIKV infection. Studies estimate that within 24 hours post-infection, fibroblasts and stromal cells undergo apoptosis (93,94). The intrinsic pathway, mediated by the release of cytochrome c in the mitochondria, activate caspase-9 and caspase-3, sub-sequentially; while extrinsic pathway seems to contribute given the several up-regulated death receptors and ligands. The increase of expression related to mitochondrial respiratory chain genes may be also related to the activation of the apoptosis intrinsic pathway found in the acute cases. The virus utilizes this cellular response to increase its rate of infection, and since the viral particles are sequestered within the apoptotic blebs, they escape recognition by the immune system. The engulfment of the apoptotic blebs by phagocytic cells, such as macrophages, can enhance infection silently (95).  Moreover, this mechanism triggers a heightened and dysregulated adaptive immune response, which may result in autoimmunity targeting self-antigens contained within apoptotic bodies (96,97), process that requires further explanations as a potential trigger for an autoimmune response. Furthermore the metabolic disturbances observed in these genes may be also associated with the exacerbate use of energy resources of host cells during cell infection (98–101).

In acute CHIKV patients, we observed an increase in the cap hypermethylation pathway, which involves the addition of the m7GMP cap structure (Cap0) to RNA. This structure is crucial for mRNA stability, processing, and translation (102–104) and is essential for viral protein synthesis. Additionally, capping helps viruses evade host immunity by preventing the recognition of terminal RNA phosphates by RIG-I and IFIT1 receptors (60,103–105). In alphaviruses, the nsP1 protein adds cap0 structures to viral RNAs (105–107), but not all viral RNAs are capped, suggesting uncapped RNAs may modulate the immune response (108,109). The increased expression of IFIT and RIG-1-like receptor genes (Supplementary Table 5) and those involved in cap hypermethylation highlights an antiviral response aimed at controlling replication and viral protein translation. Further research on RNA capping in CHIKV infections is needed to develop treatments targeting this mechanism (103). Overall, transcriptional changes during the acute phase rapidly engage immune and metabolic pathways to combat the infection.

Chronic joint involvement in Chikungunya virus infection shows similarities to rheumatoid arthritis, including shared symptoms and peripheral blood mononuclear cell profiles. Some studies attribute persistent symptoms to high levels of IL-6, GM-CSF, INF-α, and IL-17, a finding supported by our data. Other hypotheses suggest that viral RNA in the synovium and low-level replication in reservoir cells, like muscle satellite cells and synovial macrophages, drive chronic inflammation and autoimmunity (89,110,111). More recently, epigenetic changes, such as DNA hypomethylation and histone modifications, have been described in rheumatoid arthritis and other viral infections like SARS-CoV-2 (66), though these mechanisms are less explored in Chikungunya (61).

Our findings also suggest that T-cell differentiation during the transition from acute to chronic infection may play a role in sequelae. Acute infection often involves lymphopenia and a predominance of CD8+ T cells expressing CD69, CD107a, granzyme B, and perforin, targeting virus-infected cells (36,37,112,113). However, excessive antigenic exposure in the acute phase can exhaust these lymphocytes, reducing their efficacy in the chronic phase and potentially contributing to persistent symptoms (113,114). Moreover, excessive cellular infiltrates, particularly from activated memory T-cells, can expose articular tissues to cytokines and chemokines like CXCL-9, CXCL-10, CCL2, IL-6, and IL-10, leading to tissue injury and edema (114–116). While the role of T-cells in chronic Chikungunya infection remains unclear, an animal model study indicated that the absence of CD4+ T-cells reduces joint pathology (39). Additionally, key lymphocyte subsets, such as Th17 and regulatory T cells (Tregs), may be involved in chronic infection. Th17 cells promote inflammation by producing IL-17, IL-22, and IL-23, while Tregs counterbalance by releasing anti-inflammatory cytokines like IL-10 and TGF-β, easing joint symptoms. Cytokine imbalance, particularly elevated IL-1b, IL-6, and IL-17, appears to drive persistent disease, as shown in multiple studies (37,40,117,118). However, a study in Thailand during the 2009-2010 outbreak did not find a similar increase in IL-17, which could be due to methodological differences or a different CHIKV strain (119).”

Comment 10: While the paper provides important findings, the practical applications of these findings could be highlighted more. For instance, how can this knowledge inform clinical practice or future research in the field of CHIKV or other similar viruses?

Reply: We thank the reviewer for their valuable comment and the opportunity to further clarify the practical applicability of our findings.

Although the Chikungunya virus (CHIKV) poses a significant global threat, putting 3.9 billion people at risk of infection (WHO, 2024), and the chronic complications that follow infection have a profound impact on both economies and quality of life, the mechanisms underlying these chronic conditions remain poorly understood. Additionally, there is limited knowledge regarding potential therapeutic targets or pathways.

One major factor limiting advancements in this area, particularly in Brazil, is the disconnect between academia and clinical practice, especially in the context of endemic viruses like arboviruses. To address this gap, our study aimed to integrate the efforts of epidemiological surveillance and medical monitoring services into a multidisciplinary team. This approach provided additional data on potential mechanisms that contribute to the persistence of post-CHIKV symptoms and helped identify possible biomarkers in acute patients that may predict progression to chronicity.

While the high cost and complexity of RNA sequencing make it impractical for routine clinical use at this stage, our findings still hold preliminary value. The ability to discriminate early on between patients who may develop chronic symptoms and those who may not could inform future research. For instance, our results could inspire further studies involving cell culture assays to validate the molecular mechanisms associated with the gene expression profiles we identified. These mechanisms may, in turn, serve as the basis for developing new drugs aimed at modulating the inflammatory response and potentially alleviating persistent chronic pain. Unfortunately, in our study, it was not feasible to follow acute patients long enough to determine whether their symptoms resolved or persisted after the convalescent phase.

Although this study focuses specifically on the mechanisms involved in CHIKV infection, the same approach could and should be applied to other emerging and endemic diseases of global concern, such as arboviruses like Dengue. Similar research could help clarify factors associated with disease severity, warning signs, and the risk of severe outcomes such as hemorrhagic dengue. Moreover, the findings may also be relevant for understanding other viruses with pandemic potential, such as SARS-CoV-2, and even non-viral chronic diseases.

Additionally, this study highlights the potential benefits and the urgent need for stronger collaboration between academic research and clinical care. By bridging this gap, future research may lead to more effective clinical management of CHIKV infections, as well as other similar viral diseases.

Comment 11: The limitations of the study, including the relatively small sample size and the potential confounding factors (e.g., medication use, genetic factors), should be discussed more thoroughly.

Reply: We thank the reviewer for comment. In order to describe better the limitations and potential bias from this study we include the paragraph below:

Lines 564 to 571: “Even though our study highlights important candidates’ genes in the regulation and prognosis markers from different CHIKV infection phases, the small sample size may have reduced our ability to correctly identify these markers (120). Furthermore, factors associated with the virus, such as strain and viral load during the acute phase, as well as factors associated with the sample type (121) and the study population (121) may add additional bias to the analyses. Above all, the use of immunosuppressive medication by the chronic population may affect gene expression and module identification (122), acting as a potential confounding factor for reproducibility and specificity of gene set analysis.”

Round 2

Reviewer 2 Report

Comments and Suggestions for Authors

Most of the previously pointed out issues have been specifically and well revised. I think it is now appropriate for this paper to be published.